# Novel embryological classifications of hepatic arteries based on the relationship between aberrant right hepatic arteries and the middle hepatic artery: A retrospective study of contrast-enhanced computed tomography images

Mio Uraoka*, Naotake Funamizu, Kyosei Sogabe, Mikiya Shine, Masahiko Honjo, Kei Tamura, Katsunori Sakamoto, Kohei Ogawa, Yasutsugu Takada

Department of Hepatobiliary-Pancreatic Surgery and Breast Surgery, Ehime University Hospital, Shitsukawa Toon City, Ehime, Japan

* uraoka.mio.lr@ehime-u.ac.jp

## Abstract

### Background

Variations in hepatic arteries are frequently encountered during pancreatoduodenecomy. Identifying anomalies, especially the problematic aberrant right hepatic artery (aRHA), is crucial to preventing vascular-related complications. In cases where the middle hepatic artery (MHA) branches from aRHAs, their injury may lead to severe liver ischemia. Nevertheless, there has been little information on whether MHA branches from aRHAs. This study aimed to investigate the relationship between aRHAs and the MHA based on the embryological development of visceral arteries.

### Methods

This retrospective study analyzed contrast-enhanced computed tomography images of 759 patients who underwent hepatobiliary-pancreatic surgery between January 2011 and August 2022. The origin of RHAs and MHA courses were determined using three-dimensional reconstruction. All cases of aRHAs were categorized into those with or without replacement of the left hepatic artery (LHA).

### Results

Among the 759 patients, 163 (21.4%) had aRHAs. Five aRHAs patterns were identified: (Type 1) RHA from the gastroduodenal artery (2.7%), (Type 2) RHA from the superior mesenteric artery (SMA) (12.7%), (Type 3) RHA from the celiac axis (2.1%), (Type 4) common hepatic artery (CHA) from the SMA (3.5%), and (Type 5) separate branching of RHA and LHA from the CHA (0.26%). The MHA did not originate from aRHAs in Types 1–3, whereas in Type 4, it branched from either the RHA or LHA.

**Data Availability Statement:** Data cannot be shared publicly because it includes patients' personal information. Data are available from the Ehime University Data Access / Ethics Committee (contact via +81899605327) for researchers who meet the criteria for access to confidential data.

**Funding:** The authors received no specific funding for this work.

**Competing interests:** The authors have declared that no competing interests exist.

## Conclusions

Based on the developmental process of hepatic and visceral arteries, branching of the MHA from aRHAs is considered rare. However, preoperative recognition and intraoperative anatomical assessment of aRHAs is essential to avoid injury.

## Introduction

Pancreatoduodenectomy (PD) is the standard treatment for periampullary and pancreatic head tumors [1]. Since variations in hepatic arteries (HAs) are frequently observed during PD, with a prevalence of 28–46% [2–5], they are thought to increase the risk of intraoperative vascular injury. In turn, this may lead to life-threatening complications such as liver abscess or liver failure [6].

HA variations had been previously classified by Michels [7] and Hiatt et al. [8] The preoperative identification of these anomalies is essential. An aberrant right hepatic artery (aRHA) originating from the gastroduodenal artery (GDA) is a problematic variation because resection of the GDA is necessary during PD [2, 3, 9]. There are other aRHA variations such as aRHA from the superior mesenteric artery (SMA) and aRHA from the celiac axis (CA). As they are considered to have retroportal courses [4], arterial compromise could occur during lymph node dissection within the hepatoduodenal ligament.

Generally, lateral sectors (segments II and III) are supplied by the left hepatic artery (LHA), the middle hepatic artery (MHA) flows into the medial segment (segment IV), and the right lobe (segments V, VI, VII, and VIII) is supplied by the RHA. When the MHA originates from aRHAs, intraoperative injury to the aRHAs may also inhibit blood flow to the MHA, resulting in more extensive liver ischemia [10, 11]. Therefore, it is crucial to consider the intrahepatic course in aRHA cases.

Although there have been various reports and classifications on aRHAs, to the best of our knowledge, there have been no reports on the intrahepatic course of aRHAs. Therefore, this study aimed to investigate the branching of aRHA over the course of the MHA in association with the embryonic development of the HAs.

## Patients and methods

This study was approved by the ethics committee of the authors' department (approval number: 2102012). All procedures performed in studies involving human participants were conducted per the ethical standards of our institution and research committee and the 1964 Helsinki Declaration and its later amendments or comparable ethical standards. Informed consent was based on the opt-out principle and is disclosed on the study website, which included general information and provided the option to decline to participate in this study.

Preoperative contrast-enhanced computed tomography (CT) images of 759 patients who underwent hepatobiliary-pancreatic surgery, including PD, distal pancreatectomy, hepatectomy, extra bile duct resection, and extended cholecystectomy, between January 1st, 2011 and August 31st, 2022, were retrospectively reviewed. The collected data was accessed between September 1st and November 30th, 2022. Authors could access information that could identify individual participants during and after data collection. All variations of the RHA were extracted and categorized for a detailed evaluation of MHA branching morphology. To validate whether the MHA could branch from either the left or the right aberrant HA, we also

**Table 1. Dynamic liver CT scan parameters.**

| CT scan | Aquilion One (Canon Medical Systems, Otawara, Japan) | | |
|---|---|---|---|
| | SOMATOM FORCE (Siemens Healthcare AG, Erlangen, Germany) | | |
| Workstation | SYNAPSE 5 (Fujifilm Medical, Tokyo, Japan) | | |
| Contrast medium | Iopamiron 370 (Bayer Yakuin, LTD, Osaka, Japan) | | |
| The contrast dose | 600mgI/kg | | |
| Flow rate(ml/s) | 3-5mL/s | | |
| Injection duration | 30sec | | |
| tube voltage | 120kVp | | |
| **Bolus-tracking method** (determine the start of scanning in each phase after contrast medium injection) | | | |
| The trigger threshold was set at **150 Hounsfield units (HU)** | | | |
| Trigger delay | Artery phase | 15sec | |
| | Portal vein phase | 20sec | (from previous scan) |
| | Delay phase | 50sec | (from previous scan) |

focused on the anatomy of the LHA in this study. The aRHAs were classified as either with or without the replaced left hepatic artery (rLHA). Subsequently, the MHA course was obtained for each case. The configuration of the HAs was tracked by a three-dimensional (3D) CT visualization technique using Synapse Vincent® (Fujifilm, Tokyo, Japan).

## CT and 3D CT visualization

Patients in this study underwent preoperative liver-protocol CT scans using either 640-slice CT (Aquilion One; Canon Medical Systems, Otawara, Japan) or 192×2-slice CT (SOMATOM FORCE; Siemens Healthcare AG, Erlangen, Germany) scanners, with a slice thickness of 0.625–5 mm. After contrast medium injection, the arterial, portal, and delay phases were scanned in 15, 20, and 50 seconds respectively. The detailed liver-protocol CT technique is summarized in Table 1. The CT data were processed into 3D visualization images using Synapse Vincent®. The visceral arteries and the pancreas were extracted in the arterial phase. Subsequently, the portal veins were extracted in the portal vein phase. These images were fused to construct 3D images.

## Image analysis

The 3D and CT images of the arterial phase were retrospectively reviewed by two board-certified surgeons, MU and NF, with 11 and 22 years of experience in general surgery, respectively. When there was difficulty in tracking the peripheral course of HAs, the maximum intensity projection (MIP) image was constructed using Synapse Vincent®. Thereafter, all cases of HA variations were extracted.

## Standard terminology

Aberrant artery [12]: Artery with an origin or anatomical course that is substantially atypical.

Common hepatic artery (CHA) [3]: This artery supplies at least one hepatic segment and raises the GDA regardless of its origin and course.

Proper hepatic artery (PHA) [12]: Artery with the continuation of the CHA branches in the left, middle, and/or right hepatic artery.

Gastroduodenal artery (GDA) [12]: The artery originates from the CHA and courses caudally behind the first part of the duodenum anterior to the common bile duct.

Right hepatic artery (RHA) [13]: The artery originates from the PHA and divides into an anterior branch supplying segments V and VIII and a posterior branch supplying segments VI and VII. The anterior branch usually perfuses segment I and the gallbladder.

Left hepatic artery (LHA) [13]: Artery supplying segments II and III.

Middle hepatic artery (MHA) [14]: Artery in the umbilical fossa that supplies segment IV branching from the RHA, LHA, or PHA.

Replaced artery [15]: Artery receiving blood supply from an ectopic location.

Accessory artery [15]: Artery derived from both typical and ectopic branches.

Replaced RHA (rRHA) [9]: RHA originating from the SMA.

Replaced LHA (rLHA) [9]: LHA originating from the left gastric artery (LGA).

## Results

### Variations of the RHA

Over a period of 11 years, 759 patients underwent hepatobiliary-pancreatic surgery in our department. Of these, 163 (21.4%) had aRHAs. These variations were categorized into five types based on their origin (Fig 1): Type 1 included 21 cases (2.7%) of the RHA arising from GDA. Type 2 included 97 cases (12.7%) of rRHA. Type 3 included 16 cases (2.1%) of RHA arising from the CA. Type 4 included 27 cases (3.5%) of CHA from the SMA (hepatomesenteric trunk). Type 5 included two cases (0.26%) of RHA and LHA separately branching from the CHA. Among the 21 cases with aRHA from the GDA, the posterior branch arising from the GDA was observed in two patients, which might be a subtype of Type 1. The aRHAs ascended the retroportal course in Types 1, 2, 3, and 4. However, the aRHA coursed anteriorly to the common bile duct in Type 5 (Fig 2).

### Configuration of the MHA in variations of RHAs, with or without the presence of the rLHA (Table 2)

The variations in the RHAs listed above were further classified based on the presence of rLHA. Type 5 was excluded because the rLHA was absent in those two cases. The origin of the MHA was tracked with eight patterns: (Type1-a) aRHA from GDA with rLHA, (Type1-b) aRHA from GDA with normal LHA, (Type 2-a) rRHA with rLHA, (Type 2-b) rRHA with normal LHA, (Type 3-a) aRHA from CA with rLHA, (Type 3-b) aRHA from CA with normal LHA, (Type 4-a) hepatomesenteric trunk with rLHA, and (Type 4-b) hepatomesenteric trunk with normal LHA. Table 2 shows the MHA courses for each type. When the aRHA originated from the GDA, SMA, or CA (Types 1–3), in the presence of the rLHA, the MHA branched from the CHA or rLHA. The MHA branched from the normal LHA without rLHA. There were no cases of the MHA bifurcating from aRHAs in Types 1–3. In contrast, in patients with hepato-mesenteric trunk (Type 4), the MHA originated from the RHA in the presence of the rLHA, while the MHA branched from either the LHA or RHA in the absence of rLHA.

## Discussion

The main findings of this study were as follows: (1) Four patterns of RHA variations were identified. (2) The origin of the MHA could be classified with or without the rLHA. (3) In cases where the RHA originated from the GDA, CA, or SMA, the MHA was not bifurcated from the RHA. Moreover, no cases of the MHA originating from the rLHA were observed. (4) With the hepatomesenteric trunk, the MHA could branch from the RHA, LHA, and rLHA. (5) The RHA originating from the GDA, CA, and SMA exhibited a retroportal course. In contrast,

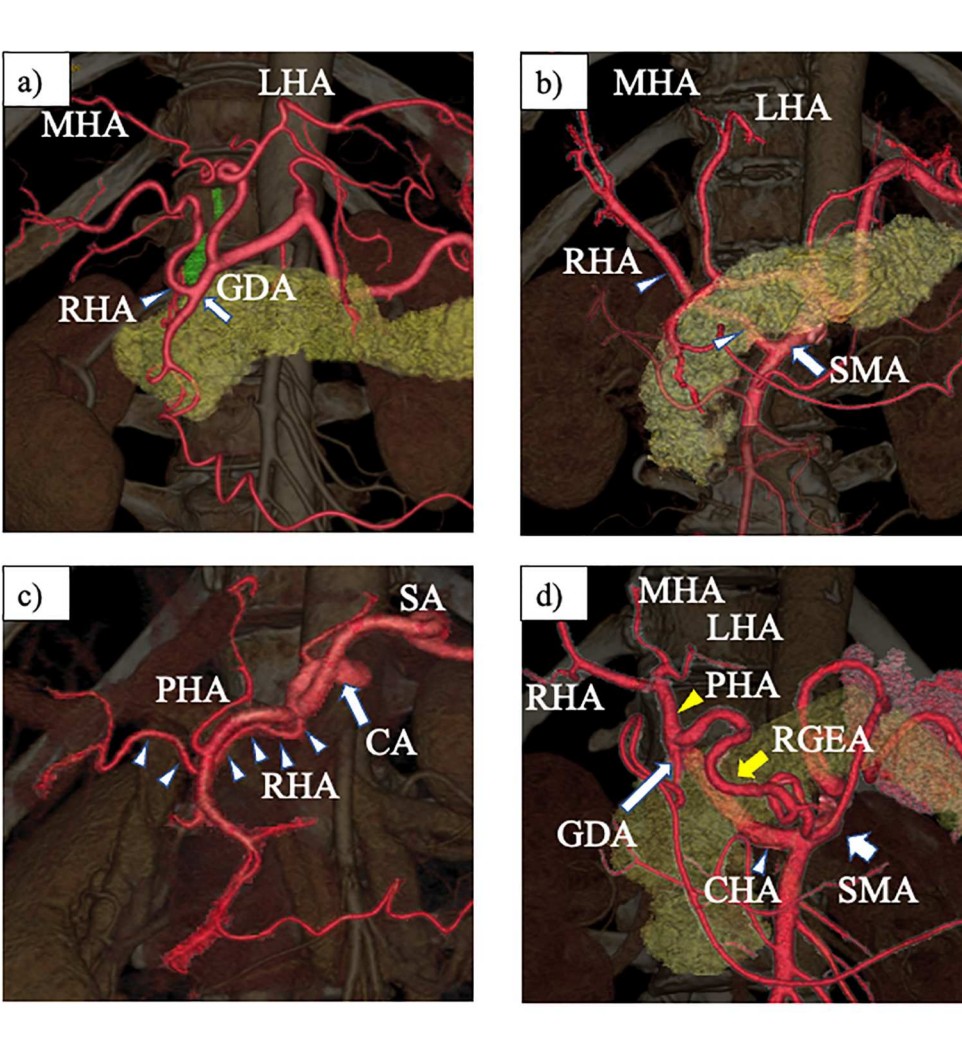

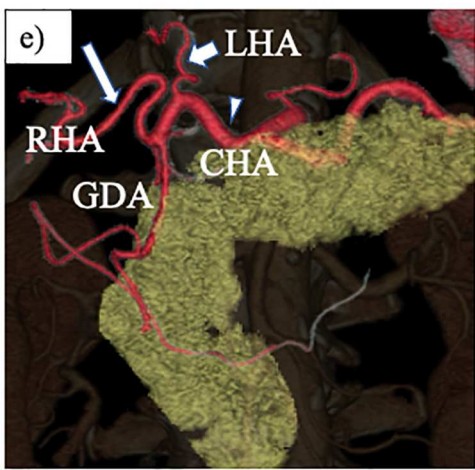

**Fig 1. VR images of patients with aberrant RHAs.** a) Type 1: RHA (arrowhead) arising from the GDA (arrow). b) Type 2: RHA (arrowhead) originating from the SMA (arrow). c) Type 3: RHA (arrowhead) branching from the CA (arrow). d) Type 4: CHA (white arrowhead) originating from the SMA (short white arrow), trifurcating GDA (white long arrow), PHA (yellow arrowhead), and RGEA (yellow arrow). e) Type 5: LHA (short arrow) and RHA (long arrow) separately originating from the CHA (arrowhead). VR, volume rendering; RHA, right hepatic artery; GDA, gastroduodenal artery; SMA, superior mesenteric artery; CA, celiac axis; CHA, common hepatic artery; PHA, proper hepatic artery; RGEA, right gastroepiploic artery; LHA, left hepatic artery.

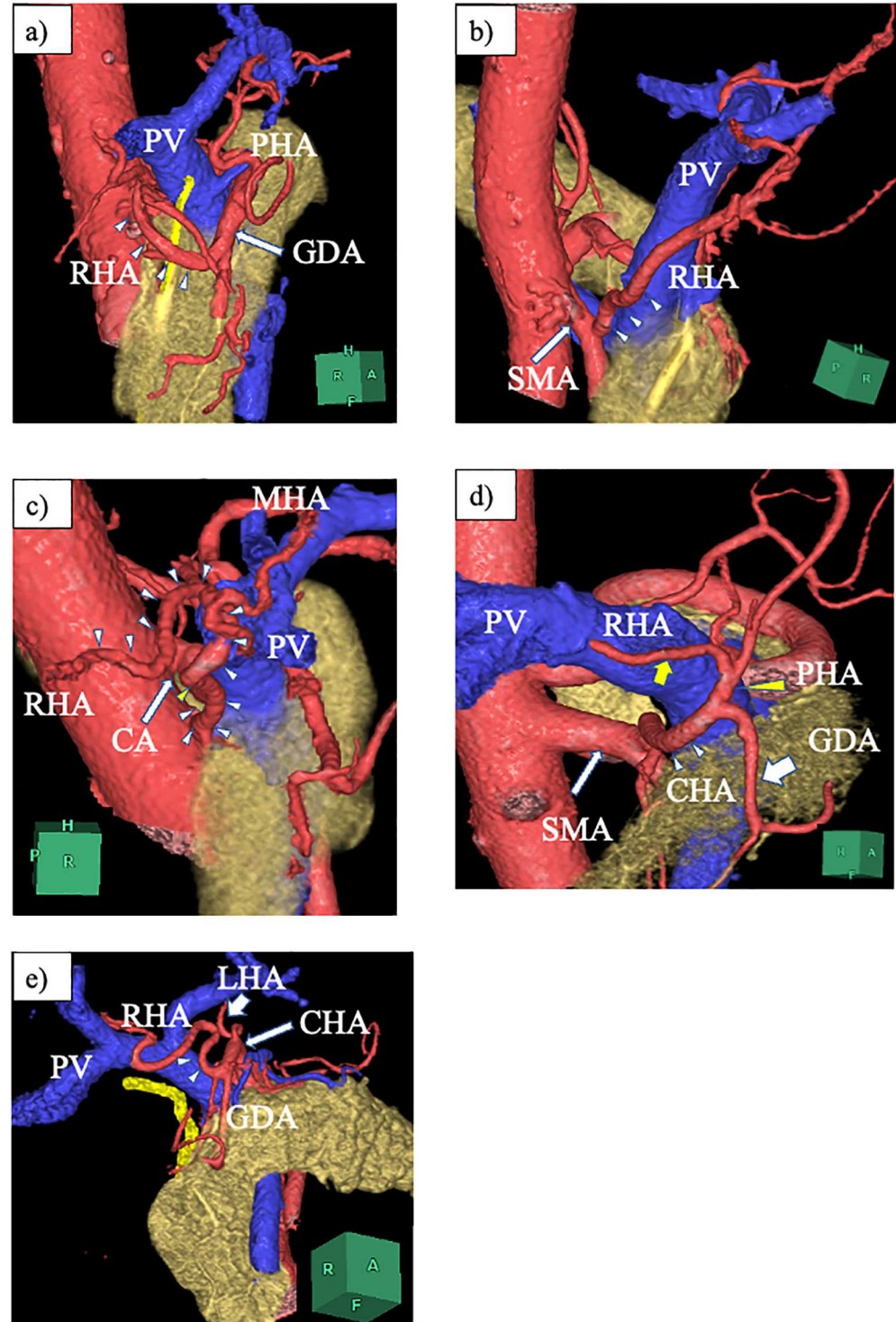

**Fig 2. Three-dimensional reconstruction images of aRHAs obtained using Synapse Vincent® (Fujifilm, Tokyo, Japan).** a) Type 1: LAO 3D view. RHA (arrowheads) ascending the retroportal way after branching from the GDA (arrow), which runs anteriorly to the pancreas. b) Type 2: LAO 3D view. After originating from the SMA (long arrow), the RHA (arrowheads) courses the dorsal side of the PV. c) Type 3: RL 3D view. RHA (starting from the yellow arrowheads and continuing along the white arrowheads) arising from the CA (arrow), which runs behind the PV. d) Type 4: LAO 3D view. CHA (white arrowheads) originating from the SMA (white long arrow) at the dorsal of the PV, then ascending just behind the PV and finally bifurcating the GDA (white short arrow) and PHA (yellow arrowhead). The RHA (yellow arrow) runs posterior to the PV. e) Type 5: LPO 3D view. LHA (short arrow) branching from the

proximal CHA (long arrow) and RHA (arrowheads) from the distal CHA. The RHA courses anterior to the PV. LAO, left anterior oblique; RL, right-left; LPO, left posterior oblique; RHA, right hepatic artery; LHA, left hepatic artery; GDA, gastroduodenal artery; SMA, superior mesenteric artery; PV, portal vein; CA, celiac axis; CHA, common hepatic artery; PHA, proper hepatic artery.

in cases where the variation that RHA and LHA originated separately from the distal CHA, the RHA followed a preportal course.

PD remains the only curative treatment for patients with pancreatic or periampullary tumors. High morbidity and mortality rates [16] are associated with the condition owing to the technical difficulty of the procedure. Injury to aRHAs because of their misidentification could result in irreversible liver ischemia, leading to severe postoperative complications, such as liver failure, liver abscess, and leakage from bilioenteric anastomosis [10, 11, 17]. Furthermore, when aRHAs branch into the MHA, the division of RHAs might be more vulnerable to ischemic damage.

Embryologically, the liver develops from three distinct lobes, each with its arterial blood supply [14, 18, 19]: The embryonic LHA from the left gastric artery perfuses segment II; the embryonic MHA from the CHA perfuses segments III, IV, V, and VIII; and the embryonic RHA from the SMA perfuses segments VI and VII (Fig 3a). The embryonic MHA courses anteriorly to the portal vein and the embryonic RHA ascends in a retroportal manner. Eventually, the embryonic LHA and embryonic RHA regress during development. However, failure of the embryonic LHA and embryonic RHA to regress results in their persistence as replaced or accessory LHA and RHA, respectively. Similarly, aRHAs may be formed through embryological arterial anastomosis and regression. During the regression of embryonic RHA, if anastomosis of embryonic RHA with the distal or proximal portion to the CHA occurs, aRHA from the GDA or CA would be formed (Fig 3a–3c). Based on the embryological structures of the MHA, embryonic RHA, and portal vein, it is natural to consider that the aRHA from the GDA, SMA, and CA courses posteriorly to the portal vein.

According to Tandler's embryological development of the visceral arterial arches, the CA arises from three primitive roots, and the SMA originates from the fourth root (Fig 3d): the LGA, splenic artery (SA), CHA, and SMA from top to bottom [20, 21]. When the longitudinal

**Table 2. aRHA classification with and without rLHA and schematic representation of the MHA origin.**

| Variations of RHA (n, %) | rLHA | |
| --- | --- | --- |
| | a: Present (n, %) | b: Absent (n, %) |
| Type 1 (RHA from GDA) n = 21, 2.7% | MHA from CHA (n = 4, 0.52%) | MHA from LHA (n = 17, 2.2%) |
| Type 2 (RHA from RHA) n = 97, 12.7% | MHA from CHA (n = 16, 2.1%) | MHA from LHA (n = 79, 10.2%) |
| | MHA from rLHA (n = 3, 0.39%) | |
| Type 3 (RHA from CA) n = 16, 2.1% | MHA from CHA (n = 3, 0.39%) | MHA from LHA (n = 13, 1.7%) |
| Type 4 (CHA from SMA) n = 27, 3.5% | MHA from PHA (n = 3, 0.39%) | MHA from LHA (n = 16, 2.1%) |
| | | MHA from RHA (n = 8, 1.1%) |

RHA: right hepatic artery, LHA: left hepatic artery, rLHA: replaced LHA, MHA: middle hepatic artery, GDA: gastroduodenal artery, CHA: common hepatic artery, PHA: proper hepatic artery, CA: celiac axis, SMA: superior mesenteric artery

anastomosis is interrupted between the third (CHA) and fourth (SMA) roots, a normal configuration of the CA and SMA is formed. Hepatomesenteric trunk may be attributed to the interruption of the longitudinal anastomosis between the second (SA) and third (CHA) roots (Fig 3e and 3f) [19].

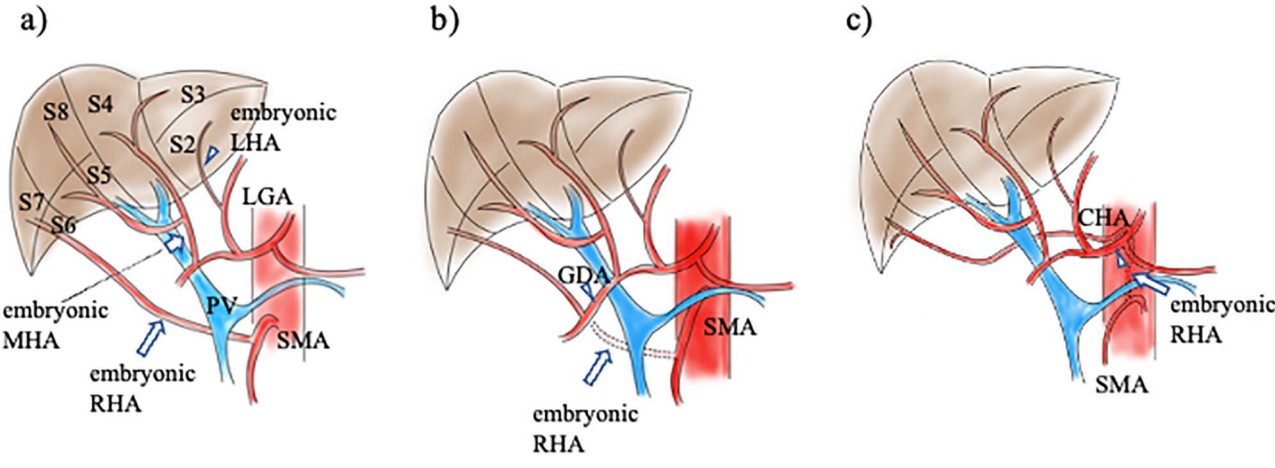

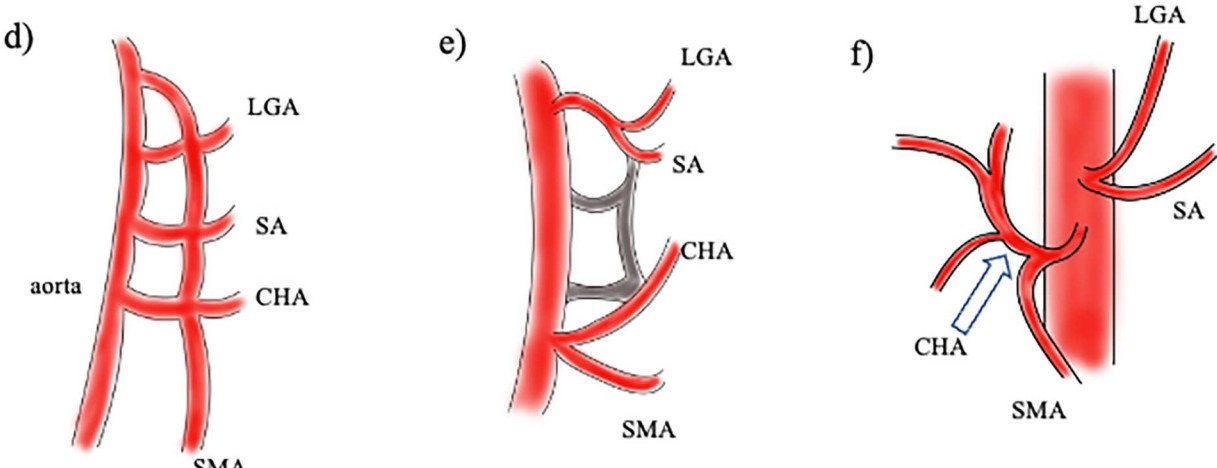

**Fig 3. The embryology of hepatic arteries and visceral arteries.** a) Three embryonic hepatic arteries and their branches with the respective Couinaud's segment in the liver. Segment II is supplied by the eLHA (arrowhead), segments III, IV, V, and VIII are supplied by the embryonic MHA (short arrow), and segments VI and VII by the embryonic RHA (long arrow). The embryonic MHA courses anterior to the PV, and the embryonic RHA ascends the retroportal way. The embryonic LHA arises from the LGA, the embryonic MHA from the CHA, and the embryonic RHA from the SMA. The embryonic RHA and embryonic LHA regress during development. b) Schematic presentation of the RHA arising from the GDA. When the anastomosis between the peripheral embryonic RHA (arrow) and GDA (arrowhead) is made, the aRHA from the GDA is formed. c) aRHA formation from the CA. When the embryonic RHA (arrow) with medial courses anastomose with the proximal portion of the CHA (arrowhead), the aRHA from the CA is formed. PV, portal vein; LGA, left gastric artery; CHA, common hepatic artery; SMA, superior mesenteric artery; GDA, gastroduodenal artery; CA, celiac axis; aRHA, aberrant hepatic artery. d) Tandler's embryological model of the visceral arteries. The CA consists of three primitive roots: the LGA, SA, and CHA. The SMA derives from the fourth root. They are joined by longitudinal anastomosis. e) The formation of hepatomesenteric trunk. When the longitudinal anastomosis between the second and the third roots is interrupted, and the third and fourth roots are subsequently joined, the hepatomesenteric trunk is formed. f) Completed form of the hepatomesenteric trunk. The CA bifurcates only the LGA and SA. The CHA arises from the SMA. CA, celiac axis; LGA, left gastric artery; SA, splenic artery; CHA, common hepatic artery; SMA, superior mesenteric artery.

Choi et al. [4] defined the term "aberrant HAs" as HAs with substantially atypical origin or anatomic course. They defined RHAs originating from the GDA and CA as "aberrant RHA." In our study, RHAs from the GDA, SMA, and CA had abnormal origins and retroportal courses. Concerning hepatomesenteric trunk, which is a variation of the visceral arterial arches, HAs would follow the normal developmental process. In addition, the variation in the RHA and LHA separately derived from the CHA was considered to be a minor anomaly in that the RHA showed a preductal course and was thought to maintain normal anatomy within the hepatoduodenal ligament. Accordingly, in this study, aRHAs were defined as Types 1–3 (Table 2). Type 4 was not considered as an aRHA.

The branching patterns of MHA were likely associated with LHA patterns in cases of aRHAs. With normal LHA anatomy, the MHA arose from the LHA; however, with rLHA, the MHA could originate from both the rLHA and CHA.

The MHA originates in the hepatic hilum and is defined as a segment IV artery. It runs outside the liver to the right of the umbilical part of the left portal vein (Fig 4) [7, 14]. Previous studies have reported that MHA originated from the RHA or LHA in approximately equal proportions [7, 22]. Wang et al. [14] reported no cases of MHA originating from an accessory hepatic artery. Conversely, Xie et al. [23] noted that there were three cases (0.6%) of the MHA originating from an rRHA and 15 cases (3.3%) from rLHA. There was no case of MHA branching from rRHA in this study.

The structure of the embryonic MHA was the basis for the MHA, which was thought to be responsible for most of the liver perfusion after development. During the regression of the roots of the embryonic LHA and embryonic RHA, intrahepatic communication among the peripheral branches of the embryonic LHA, embryonic MHA, and embryonic RHA was performed. Therefore, the MHA could not have branched from the proximal embryonic LHA or embryonic RHA, which was originally destined to regress. Both in the previous report by Xie et al. [23] and our analysis, there was a small number of cases of the MHA derived from an

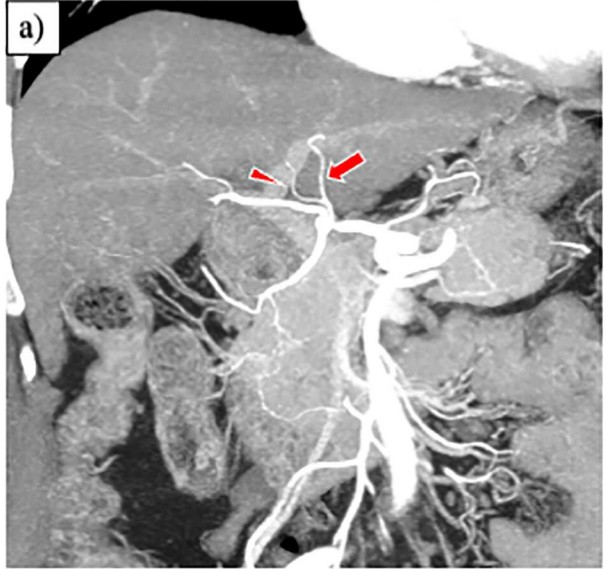
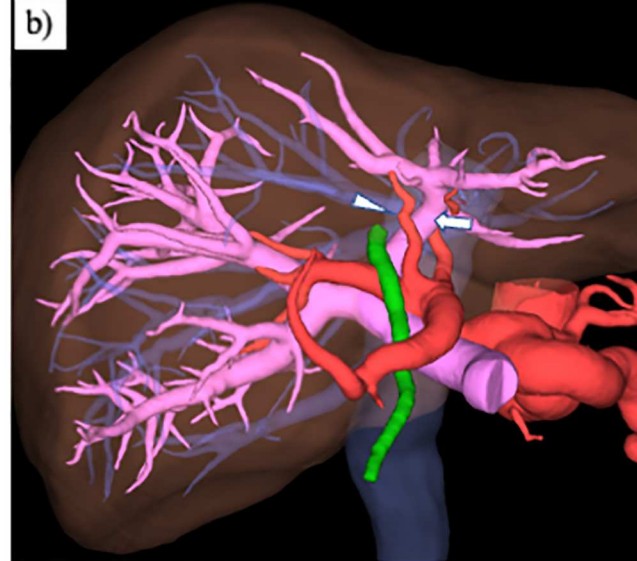

**Fig 4. Normal anatomy of HAs.** a) Oblique coronal MIP image. The MHA (arrowhead) arises from the first branch of the LHA (arrow) at the hepatic hilum. b) Three-dimensional reconstruction image using Synapse Vincent® (Fujifilm, Tokyo, Japan). The MHA (arrowhead) courses outside the liver to the right of the umbilical part of the left portal vein (arrow). MIP, maximum intensity projection; HA, hepatic artery; MHA, middle hepatic artery; LHA, left hepatic artery.

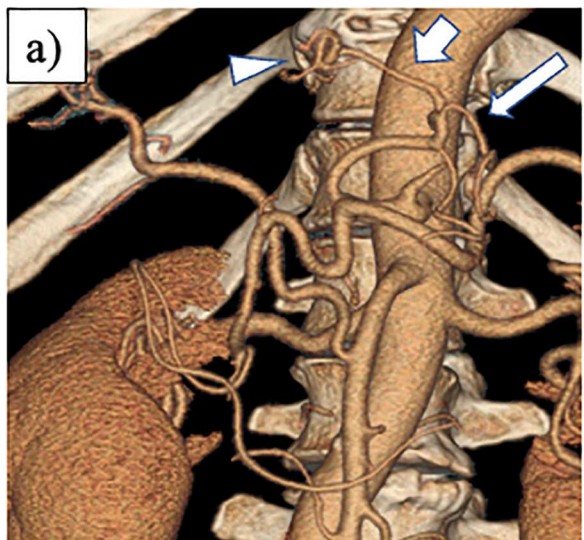

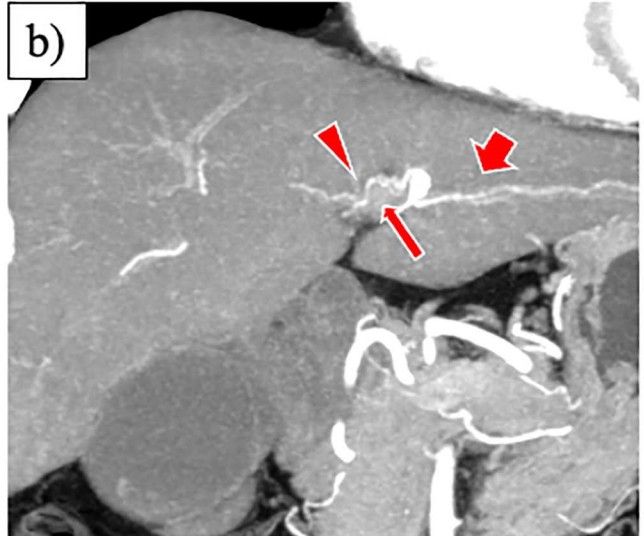

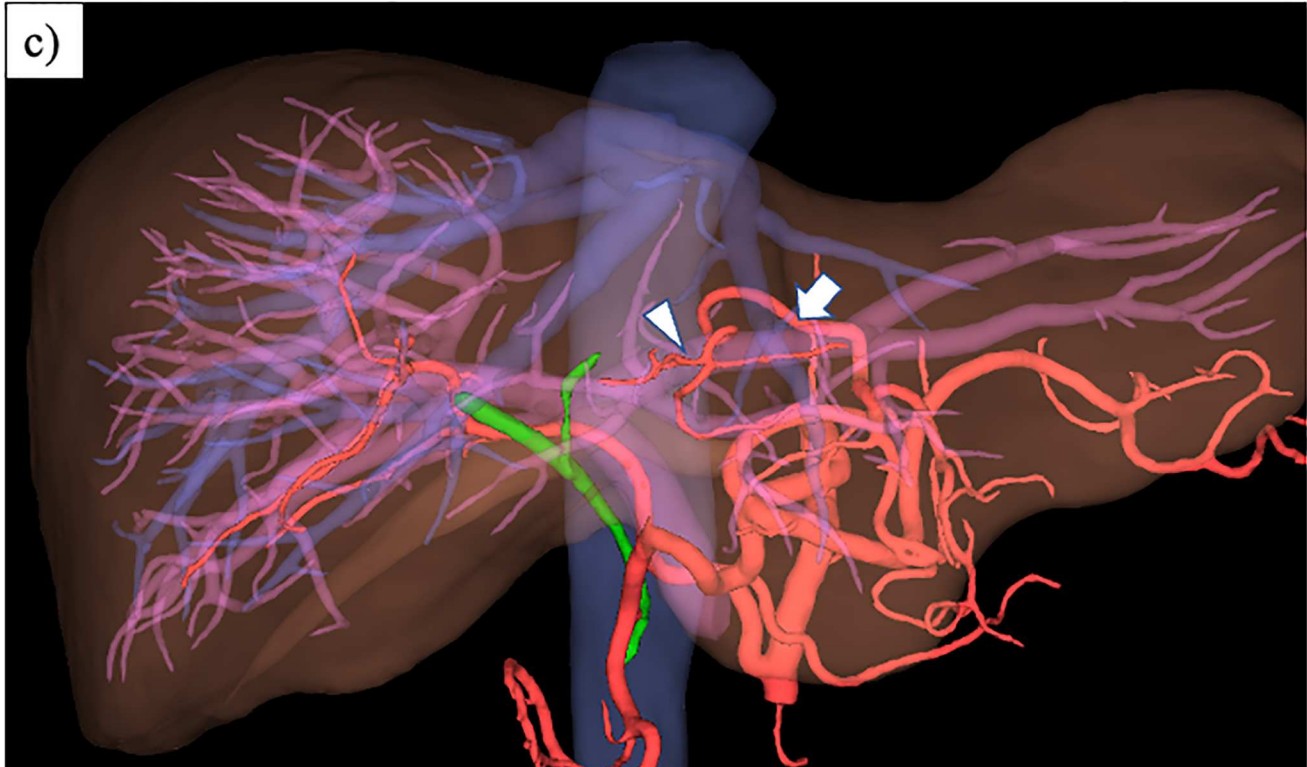

**Fig 5. The artery to hepatic segment IV originating from the left hepatic artery inside the umbilical fissure.** a) VR image of the RHA arising from the SMA with the rLHA. The rLHA (short arrow) originates from the LGA (long arrow) and then branches the MHA (arrowhead). b) Oblique coronal MIP image showing how the MHA (arrowhead) originates from the rLHA inside the umbilical fissure (arrow). c) Three-dimensional reconstruction image using Synapse Vincent® (Fujifilm, Tokyo). The MHA (arrowhead) courses inside the umbilical fissure after branching from the rLHA (arrow). VR, volume rendering; MIP, maximum intensity projection; RHA, right hepatic artery; MHA, middle hepatic artery; LHA, left hepatic artery; rLHA, replaced LHA; LGA, left gastric artery; SMA, superior mesenteric artery.

rLHA or rRHA. It has been mentioned that there is an artery to hepatic segment IV originating from the LHA inside the umbilical fissure (Fig 5) [14] Considering that the MHA and LHA distribute the left lobe perfusion, it might be natural that communication occurs between the arteries to segment IV and the peripheral branch of the rLHA inside the umbilical fissure. Because cases of the MHA from an rRHA are extremely rare, further embryological investigation of this variation is required.

Although resection of aRHAs during PD remains controversial [24, 25], radical resection can be achieved whenever possible [26]. Preoperative strategies, such as embolization, dissection, preservation, and reconstruction, are required to manage aRHAs. When aRHAs are accidentally or necessarily sacrificed, portal vein and intrahepatic collateral flow usually compensate for liver perfusion [9, 27]. Yamamoto et al. [10] radiologically identified that arterial flow through the hepatic and hepatoduodenal ligaments could be collateral pathways immediately after artery embolization. However, it is widely known that the loss of arterial flow causes biliary fistula [10, 11, 17]. The blood supply to the proximal bile duct mainly depends on the RHA [9, 24]. Therefore, sufficient RHA flow is required to prevent biliary complications.

Preoperative and intraoperative evaluations of arterial anatomy are strongly recommended [28]. Recognition of HA configurations and the relationship between the tumor and aRHAs would be more accessible through the 3D description of arteries and the pancreas. For the intrahepatic assessment of aRHAs, observing the anatomy of aRHAs using ultrasonography could be helpful.

Our study has several limitations. First, compared with cadaveric studies or digital subtraction angiography, image analysis by contrast-enhanced CT might be inferior in depicting small arteries. However, 3D visualization of CT should provide an accurate and precise arterial anatomy comparable to these modalities. Second, the number of patients included in this study was smaller than that in previous large-series arterial anatomy studies. A more accurate classification of the branching patterns of the MHA from aRHAs could be possible by collecting more cases.

In conclusion, the results of our study indicated that the MHA did not branch from aRHAs. However, since there have been some reports on the MHA originating from aRHAs, it is necessary to pay attention to the course of MHAs. Furthermore, intraoperative maneuvers such as lymph node dissection within the hepatoduodenal ligament should be carefully performed since aRHAs follow a retroportal course. When aRHA resection is required during PD, liver perfusion is relatively preserved since the MHA and LHA compensate for the toll on liver flow. The association between aRHAs and the MHA from an embryological viewpoint was confirmed in the present study. Preoperative simulation of aRHAs and their intrahepatic course by 3D reconstruction and intraoperative assessment of the course of aRHA using ultrasonography are necessary to preserve aRHAs during PD.

## Acknowledgments

The authors thank radiologic technologists Taichi Furumochi and Hiroshi Suekuni from the Department of Radiology of the Ehime University Graduate School of Medicine, for their support and advice on 3D CT imaging methods.

## Author Contributions

**Supervision:** Naotake Funamizu, Yasutsugu Takada.

**Writing – original draft:** Mio Uraoka.

**Writing – review & editing:** Naotake Funamizu, Kyosei Sogabe, Mikiya Shine, Masahiko Honjo, Kei Tamura, Katsunori Sakamoto, Kohei Ogawa.

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
