## [Decision Letter · Decision Letter 0]

20 Nov 2023

PONE-D-23-27540Novel embryological classifications of hepatic arteries based on the relationship between aberrant right hepatic arteries and the middle hepatic artery: a retrospective study of contrast-enhanced computed tomography imagesPLOS ONE

Dear Dr. Uraoka,

Thank you for submitting your manuscript to PLOS ONE. After careful consideration, we feel that it has merit but does not fully meet PLOS ONE’s publication criteria as it currently stands. Therefore, we invite you to submit a revised version of the manuscript that addresses the points raised during the review process.

Reviewers provided several important questions.

Please respond to these questions, or if you are unable to respond, please address these issues in the discussion section. In particular, I think it would be better to describe the clinical impact of this study more clearly.

We look forward to receiving your revised manuscript.

Kind regards,

Yoshihisa Tsuji

Academic Editor

PLOS ONE

4. We note that Figures 1, 2, 3, 4, and 5 in your submission contain copyrighted images. All PLOS content is published under the Creative Commons Attribution License (CC BY 4.0), which means that the manuscript, images, and Supporting Information files will be freely available online, and any third party is permitted to access, download, copy, distribute, and use these materials in any way, even commercially, with proper attribution. For more information, see our copyright guidelines: http://journals.plos.org/plosone/s/licenses-and-copyright.

1. You may seek permission from the original copyright holder of Figures 1, 2, 3, 4, and 5 to publish the content specifically under the CC BY 4.0 license.

5. Please upload a copy of Figure 6, to which you refer in your text on page 22. If the figure is no longer to be included as part of the submission please remove all reference to it within the text.

Reviewers' comments:

Reviewer's Responses to Questions

**Comments to the Author**

1. Is the manuscript technically sound, and do the data support the conclusions?

Reviewer #1: Partly

Reviewer #2: Partly

2. Has the statistical analysis been performed appropriately and rigorously? 

Reviewer #1: N/A

Reviewer #2: Yes

3. Have the authors made all data underlying the findings in their manuscript fully available?

Reviewer #1: Yes

Reviewer #2: Yes

4. Is the manuscript presented in an intelligible fashion and written in standard English?

Reviewer #1: No

Reviewer #2: Yes

5. Review Comments to the Author

Reviewer #1: The authors have examined the anatomical patterns of hepatic arteries and have identified that MHA can branch from aRHA only when CHA branches from SMA (Type 4). However, there is a lack of logical consistency, and the clinical significance of their findings remains unclear.

<major issues="">

1. I think that an injury to the aRHA can result in severe liver ischemia, regardless of the origin of the MHA. Additionally, while the authors described “little is known about whether the MHA branches from aRHAs”, they also mentioned that such anomaly can be a cause of severe liver ischemia due to intraoperative injury. They should clarify what is unknown and why they are investigating this topic.

2. There is no explanation provided as to why the authors classified the anatomical patterns of hepatic arteries based on the presence or absence of rLHA, not only the RHA and MHA patterns. They should specify why the information about rLHA was necessary for this study?

3. The most significant finding of this study is that the authors found MHA branching from aRHA only in Type 4b. However, whether MHA branches from RHA or LHA in Type 4 appears to have little impact on the risk of severe liver ischemia during PD. I believe that recognizing Type 1a and Type 3 is more important for avoiding intraoperative injury to MHA.

4. Much of the information in the discussion section repeats what was described in the introduction section. The authors should provide a more concise discussion of what they have learned from this study, how it differs from previous reports, and how their findings can be applied to improve clinical practice.

5. Their conclusion that "preoperative recognition and intraoperative assessment of aRHA flow is essential to avoid injury" cannot be supported by their findings.</major>

Reviewer #2: “Novel embryological classifications of hepatic arteries based on the relationship between aberrant right hepatic arteries and the middle hepatic artery: a retrospective study of contrast-enhanced computed tomography images” by Urakawa et al. describes the classification of hepatic arteries based on the data of the origin of RHAs and MHA courses. It is a important clinical concern, and the manuscript is well written. However, there are some issues to be addressed as following;

Major

1 This is only a classification, and the clinical utility should be demonstrated. As there have been several classifications of hepatic arteries previously, authors should compare them and their own classifications in the point of view of clinical importance, and demonstrate the superiority of their classification.

2 They describe the frequencies of each variant of hepatic arteries, however, these are not validated elsewhere, especially, several rare variants described here.　

3 They conclude that preoperative recognition and intraoperative assessment of aRHA flow is essential to avoid injury. There is no data in this manuscript supporting this conclusion. It is important that surgery-associated injury can be significantly reduced when their novel classification is recognized by surgeons. If they have experienced any cases in which unexpected injury occurred during hepatobiliary-pancreatic surgery without the knowledge of the novel classifications of hepatic arteries, they might be helpful for demonstrating the necessity of their novel classification.

6. PLOS authors have the option to publish the peer review history of their article (what does this mean?). If published, this will include your full peer review and any attached files.

Reviewer #1: No

Reviewer #2: No

---

## [Author Response · Author response to Decision Letter 0]

26 Dec 2023

Response to Reviewers

We sincerely appreciate the time and effort you have dedicated to reviewing our manuscript. Your insightful comments and suggestions have been invaluable in enhancing the quality of our work. We are truly grateful for your constructive feedback.

<To Reviewer #1>

1. We added a statement emphasizing that there is currently no published paper demonstrating the relationship between variant hepatic arteries and MHA to underscore the significance of our research findings.

2. To validate that the MHA did not branch from the variant LHA, we also needed to prove that it did not branch from the replaced LHA. Therefore, we categorized the cases into two groups: those with replaced LHA and those without.

3. As mentioned in the “Discussion” section, we considered Type 4 as an overall variation of the hepatic artery rather than an aRHA. We have now added a clarification that Types 1-3 are defined as aRHA.

4. While our study may lack impact as it primarily derives its findings from embryological perspectives suggesting that the MHA does not branch from the aRHA, we believe it serves as an enlightening reminder. Surgeons should always be cautious about the detailed course of variant hepatic arteries during surgery.

5. I have removed the section regarding the clamp test.

<To Reviewer #2>

1. (For the comments 1+3) While our study may lack impact as it primarily derives its findings from embryological perspectives suggesting that the MHA does not branch from the aRHA, we believe it serves as an enlightening reminder. Surgeons should always be cautious about the detailed course of variant hepatic arteries during surgery.

2. The frequency of RHA for Types 1-4 has been widely reported in the past, and in this study, the frequency of each variant did not significantly differ from previous reports (i..e. Referrence 2,7,8).

3. I have removed the section regarding the clamp test.

---

## [Decision Letter · Decision Letter 1]

29 Jan 2024

PONE-D-23-27540R1Novel embryological classifications of hepatic arteries based on the relationship between aberrant right hepatic arteries and the middle hepatic artery: a retrospective study of contrast-enhanced computed tomography imagesPLOS ONE

Dear Dr. Uraoka,

Thank you for submitting your manuscript to PLOS ONE. After careful consideration, we feel that it has merit but does not fully meet PLOS ONE’s publication criteria as it currently stands. Therefore, we invite you to submit a revised version of the manuscript that addresses the points raised during the review process.

We look forward to receiving your revised manuscript.

Kind regards,

Yoshihisa Tsuji

Academic Editor

PLOS ONE

Journal Requirements:

Additional Editor Comments:

As results of assessment by several reviewers, the content seems to be improved compared to the previous version. Please respond to a few more comments from additional reviewers.

Reviewers' comments:

Reviewer's Responses to Questions

**Comments to the Author**

1. If the authors have adequately addressed your comments raised in a previous round of review and you feel that this manuscript is now acceptable for publication, you may indicate that here to bypass the “Comments to the Author” section, enter your conflict of interest statement in the “Confidential to Editor” section, and submit your "Accept" recommendation.

Reviewer #3: All comments have been addressed

Reviewer #4: All comments have been addressed

2. Is the manuscript technically sound, and do the data support the conclusions?

Reviewer #3: Yes

Reviewer #4: Yes

3. Has the statistical analysis been performed appropriately and rigorously? 

Reviewer #3: Yes

Reviewer #4: N/A

4. Have the authors made all data underlying the findings in their manuscript fully available?

Reviewer #3: Yes

Reviewer #4: Yes

5. Is the manuscript presented in an intelligible fashion and written in standard English?

Reviewer #3: Yes

Reviewer #4: Yes

6. Review Comments to the Author

Reviewer #3: They found branching of the middle hepatic artery from aberrant hepatic artery (aRHA) is very rare. Authors responded to the reviewers’ comment properly. This is useful for surgeons.

Reviewer #4: Uraoka et al. retrospectively reported the relationship between aRHAs and the MHA based on the embryological development of visceral arteries in 759 patients at a single institution. Two board-certified surgeons, but not radiologists, reviewed 3D and CT images of the arterial phase, and clarified the branching type of MHA. Subsequently, they concluded that branching of the MHA from aRHAs was considered rare, and however, preoperative recognition and intraoperative assessment of aRHA flow was essential to avoid injury.

The first revise process made this manuscript better.

There are a few comments to be addressed.

1. In conclusion, the authors described that branching of the MHA from aRHA was considered rare. Actual proportion of the MHA from aRHA is coincident with incidence of type 4 (2.1%)? Please clarify this proportion in the results of abstract and text.

2. There were too many abbreviations in the text. Unfamiliar abbreviation should not be used such as HMT.

3. Abbreviation in figure title should not be also used.

I appreciate for giving me an opportunity to review this manuscript.

7. PLOS authors have the option to publish the peer review history of their article (what does this mean?). If published, this will include your full peer review and any attached files.

Reviewer #3: No

Reviewer #4: No

---

## [Author Response · Author response to Decision Letter 1]

6 Feb 2024

Response to Reviewers

We sincerely appreciate the time and effort you have dedicated to reviewing our manuscript. Your insightful comments and suggestions have been invaluable in enhancing the quality of our work. We are truly grateful for your constructive feedback.

-There was an error in the values presented in Table 2. The number and percentage of patients with and without rLHA in Type 1 anomaly were mistakenly reversed. (Prior to correction: Type 1 cases with rLHA, MHA arose from CHA in 17 patients<2.2%>; without rLHA, MHA arose from LHA in four patients <0.52%>.) This has been rectified. We apologize for any confusion.

<To Reviewer #4>

1. In this paper, the aberrant RHA (aRHA) is defined as Type 1-3 since the term “aberrant artery” represents the artery with substantially atypical origin or anatomic course. Therefore, Type 4 (Hepatomesenteric trunk) was excluded from aRHAs as it was considered an anomaly of the visceral arterial arches during embryological development. Hepatic arteries in Type 4 are presumed to exhibit the typical branching pattern of ordinary hepatic arteries, MHA is thought to be capable of branching from either RHA or LHA. (We marked up the relevant section in blue, pages 18-19.)

2. We have corrected all the abbreviations, such as HMT, eLHA, eMHA, and eRHA.

3. Similarly, I have also revised those abbreviations within the figures and tables.

Finally, we sincerely thank you again for your insightful feedback.

---

## [Editor Report · Decision Letter 2]

8 Feb 2024

Novel embryological classifications of hepatic arteries based on the relationship between aberrant right hepatic arteries and the middle hepatic artery: a retrospective study of contrast-enhanced computed tomography images

PONE-D-23-27540R2

Dear Dr. Uraoka,

We’re pleased to inform you that your manuscript has been judged scientifically suitable for publication and will be formally accepted for publication once it meets all outstanding technical requirements.

Kind regards,

Yoshihisa Tsuji

Academic Editor

PLOS ONE
---

## [Editor Report · Acceptance letter]

18 Feb 2024

PONE-D-23-27540R2 

PLOS ONE

Dear Dr. Uraoka, 

I'm pleased to inform you that your manuscript has been deemed suitable for publication in PLOS ONE. Congratulations! Your manuscript is now being handed over to our production team.

Kind regards, 

on behalf of

Professor Yoshihisa Tsuji 

Academic Editor

PLOS ONE